# Assessing the school food environment and its role on healthy eating behaviours among school age children in Dar es Salaam, Tanzania

**Nyabasi Makori**[1]*, **Dyness Kejo**[1], **Hoyce Mshida**[1], **Beatrice Bachwenkizi**[1], **Devotha Mushumbusi**[1], **Zahara Daudi**[1], **Monica Chipungahelo**[1], **Ai Zhao**[2], **Anselm P. Moshi**[1]

**1** Tanzania Food and Nutrition Centre, Dar es salaam, Tanzania, **2** Vanke School of Public Health, Tsinghua University, Beijing, China

* nmakori01@gmail.com

## Abstract

School food environment plays a crucial role in shaping children's dietary habits and promoting healthy eating practices. The study investigated the school food environment in Temeke Municipality, Dar es Salaam, focusing on its role in promoting healthy eating among school-age children. A cross-sectional survey was conducted across four schools, with food vendors (N = 20) and teachers (N = 8) interviewed using structured questionnaires. The study aimed to assess food offerings, school oversight, and vendors' knowledge of food quality. The findings disclosed that 62.5% of the schools partially implemented school feeding guidelines, while 37.5% did not implement them. Among the surveyed schools, 37.5% had food storage facilities, 25.0% had functioning kitchens, and none had dining halls. The food environment included both healthy and unhealthy options, with 55.6% of food and beverages classified as healthy and 44.4% as unhealthy. Popular unhealthy items included samosas (95.5%), fried potato chips (87.0%), and fried mashed potato balls (73.9%). The study also compared the calorie portions of foods purchased by students with the Recommended Dietary Allowances (RDA) for different age groups. The percentage of RDA covered by these portions ranged from 8.0–19.0% for ages 5–8, 6.0–19.0% for ages 9–13, and 5.0–16.0% for ages 14–18. Furthermore, food vendors demonstrated low knowledge of food safety, hygiene, and nutrition, with only 22.0% aware of the national food guidelines. The overall food environment exposed students to unhealthy food options, with significant gaps in the implementation of health guidelines. Findings highlight the need for interventions to improve food offerings and promote healthier food choices around schools.

**Data availability statement:** Data is available from figshare at https://doi.org/10.6084/m9.figshare.29290379.v1.

**Funding:** This study was conducted with support from a grant funded by the Research Fund of the Vanke School of Public Health at Tsinghua University - Bright future grant The grant number award is 202330046. Funder had a role in reviewing the manuscript.

**Competing interests:** The authors have declared that no competing interest exist.

## 1. Introduction

The school food environment encompasses the various physical, economic, political, and sociocultural factors that influence children's food acquisition, purchasing behaviors, and consumption patterns within the school setting [1]. It includes the conditions, opportunities, and determinants that shape dietary choices and nutritional status among school-aged children [2,3]. The school food environment plays a pivotal role in shaping children's eating behaviors and overall health, with research indicating that the food available within and around schools significantly impacts both short-term health outcomes and long-term nutrition trajectories [4,5].

Often, schools are situated in areas with abundant food outlets offering energy-dense, low-cost foods, increasing the likelihood of students purchasing those foods. These environmental conditions directly influence students' food choices, with implications for dietary behaviors [6].

Schools are a critical setting for nutrition interventions, as children and adolescents spend a significant portion of their waking hours within the school environment, making it an ideal location for fostering healthy eating habits. The over availability of unhealthy food options in this environment may contribute to the development of poor dietary patterns, limiting acceptance of nutrient-dense foods, and potentially influencing lifelong eating habits. Early childhood nutrition is essential for promoting optimal growth, health, and cognitive development, and for mitigating both immediate health concerns and the risk of chronic diseases later in life.

Globally, malnutrition in all its forms remains a major public health challenge, affecting children, adolescents, and adults alike [3]. It is not only a health issue but also a barrier to economic development, productivity, and the achievement of global poverty reduction goals. A recent cross-sectional study in Tanzania (2024) documented a triple burden of malnutrition among school-aged children, with stunting, anemia, and overweight affecting 32%, 34%, and 9.4% of the population, respectively [7]. This trend highlights the growing vulnerability of school-aged children to malnutrition, exacerbated by shifts in food environments, inadequate integration of nutrition into health programs, and insufficient nutrition interventions.

Changes in the food environment have facilitated the widespread consumption of energy-dense foods, high in fats, sugars, and salt, as well as a marked increase in the consumption of sugar-sweetened beverages, particularly among adolescents [8]. Over the past two decades, Tanzania, like many other African countries, has witnessed a surge in demand for processed and convenience foods, driven by shifts in consumer lifestyles and time use patterns [9]. Factors such as food availability, affordability, accessibility, and desirability have become key determinants of food choices. Additionally, evolving consumer preferences, influenced by a growing middle urban class with disposable incomes, demographic transitions, and globalization, have further contributed to this dietary shift [9]. Although this trend is most pronounced in urban and peri-urban areas, evidence suggests that the dietary transition is also reaching rural communities, largely driven by changes in labor dynamics and time-saving considerations related to food preparation [10].

The shifts in food environments are especially concerning for school-aged children and adolescents, as they directly impact food preferences, dietary behaviors, and ultimately, nutritional outcomes. However, research on the school food environment and its impact on nutrition outcomes in Tanzania remains limited. To address these challenges, the Tanzanian government, through the Ministry of Education, developed the School Feeding and Nutrition Guideline in 2018. This guideline aims to provide a framework for the sustainable provision of food and nutrition services in schools, offering standardized procedures for effective implementation and ensuring consistent food availability in schools [11]. Moreover, these policies are critical in enhancing the quality of school feeding programs and ensuring their long-term sustainability at the national level [12].

However, failing to comply with the guidelines can result in inconsistent implementation of school feeding programs, which may negatively affect the quality and nutritional adequacy of the meals provided. Inadequate implementation may result in improper meal planning, which, in turn, can lead to imbalanced diets that fail to meet the nutritional needs of students, potentially hindering growth and development. Challenges in the implementation of school feeding programs have been documented in other contexts, such as Rwanda, where insufficient infrastructure has been identified as a major barrier [13]. Adherence to the basic requirements outlined in the school feeding guidelines is essential for the successful operation of these programs. These requirements include the establishment of appropriate infrastructure, such as well-ventilated kitchens equipped with fuel-efficient stoves, proper storage facilities, a reliable water supply, cooking and serving utensils, a canteen, garbage disposal systems, and adequate hygiene and sanitation facilities. The objective of this study is to assess the school food environment in Temeke District, Tanzania by examining the types of foods available, the factors influencing students' food preferences, and the role of infrastructure, vendor knowledge, and food healthiness in shaping these preferences.

## 2. Methods and approaches

### 2.1. Study area

The study was conducted in Temeke Municipal, located in the southern part of the Dar es Salaam Region, Tanzania. The municipality consists of two divisions (Mbagala and Chang'ombe), 30 wards, and 142 subwards/streets. Temeke municipality was purposefully selected because it includes low, medium, and high-income earners, as well as both rural and urban residents, making it a representative sample of the broader Dar es Salaam population. Temeke District comprises 134 primary schools, of which 51 are private institutions, and 2 cater to children with special educational needs. Additionally, the district has 63 secondary schools, 37 of which are private, and 26 are public.

### 2.2. Study design and target population

A cross-sectional survey was conducted in December 2023 to collect baseline data on the school food environment in Temeke Municipal. The survey targeted a purposively selected sample of primary and secondary schools, including both urban and peri-urban schools, to ensure representation of diverse student populations from different food environments, socio-economic backgrounds, cultural contexts, and ethnic groups.

### 2.3. Target population

Schoolchildren and adolescents aged 6–18 years from either a secondary or primary school in Temeke municipality, Dar-es salaam, Tanzania.

### 2.4. Data collection

The data collection tools were pre-tested with a small sample of food vendors to ensure the clarity, accuracy, and reliability of the instruments. Based on the feedback received during the pre-testing phase, the tools were refined to enhance

their precision and ensure they effectively captured the required information. Data collection was conducted by experienced researchers using pre-tested semi-structured questionnaires, checklists, and observation guides. These tools were designed to provide comprehensive insights into the school food environment and the factors influencing students' food choices and nutritional outcomes.

Face-to-face interviews were conducted with food vendors to gather data on the school food environment including the foods sold to students, the food products available on school premises, food pricing, students' food preferences, food safety and hygiene practices, and the vendors' knowledge of nutrition and its relevance to health. Interviews were conducted with head teachers and health teachers to explore the implementation of school feeding programs, the availability of food around the school, and other nutrition-related policies and practices. Using a structured checklist, researchers assessed various aspects of the food environment. The checklist focused on evaluating the variety and quality of food products sold, the accessibility of food outlets for students, and the types of offerings available.

### 2.5. Data analysis

The Statistical Package for the Social Sciences (SPSS) for Windows software (IBM version 21, Armonk, NY, USA) was used to analyze the information collected from food vendors and teachers. Descriptive statistical measures were employed to analyze the quantitative data, which were then presented in Tables and figures.

### 2.6. Ethical clearance

Permission to conduct the study was obtained from the National Institute for Medical Research. The Prime Minister's Office, Regional Authority, and Local Government (PORALG) granted permission to carry out the study in primary and secondary schools. The Ministry of Education, through the regional and district health and education offices in Temeke Municipal Council, informed head teachers about the upcoming research. Prior to data collection, the researchers conducted pre-visits to each school to introduce the study objectives to the head teachers. Written informed consent forms were obtained from both teachers and food vendors.

## 3. Results and discussion

### 3.1. School feeding programme, policies and guidelines

Awareness of the existing policies and guidelines related to the school feeding program was generally high across all participating schools. However, only 62.5% of the schools were partially adhering to the guidelines, while 37.5% were not implementing them at all. The survey results revealed significant deficiencies in the infrastructure necessary for the effective implementation of school feeding programs. Specifically, only 37.5% of the surveyed schools had access to a food storage facility, and merely 25% had a functional kitchen. Notably, none of the schools were equipped with a dining hall (Fig 1), which is critical for providing a conducive environment for students to consume meals and engage in social interaction.

The lack of adequate infrastructure poses a substantial challenge to the program's effectiveness. In the absence of proper storage facilities, food safety may be compromised, increasing the risk of health hazards for students. Effective food storage is essential for preserving food quality and ensuring safety; without such provisions, students may be exposed to unsafe meals that could negatively impact their health. Moreover, the absence of proper kitchen facilities can result in inefficient or unsafe food preparation, further diminishing the quality and nutritional value of the meals provided.

### 3.2. Availability of food and beverages around the school environment

The availability of food around schools can significantly impact the health and nutrition outcomes of students. Typically, schools are surrounded by shops, grocery stores, canteens, restaurants, and fast-food outlets, which can shape the types

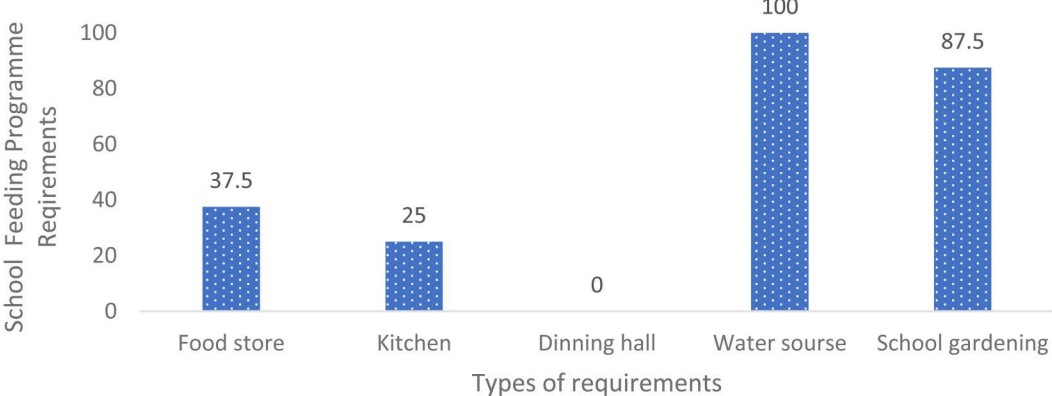

**Fig 1. Existing infrastructure to support school feeding program in primary and secondary schools.**

of food students consume. The survey findings revealed that variety of food and beverages were available around the school environment (Table 1), the majority were found to be sold at a cheaper price ranging from TZS 100–1000 (USD 0.0004–0.4) (Table 4). The availability of cheaper food options in the school environment significantly contributes to the increased consumption of unhealthy foods among students. Fast food outlets, grocery and canteens often offer low-cost snacks and meals that are high in sugar, fat, and salt, making them more appealing to students seeking quick and afford-able options. This easy access to unhealthy choices can overshadow healthier alternatives, leading to poor dietary habits [2] that may persist into adulthood. Additionally, the influence of peer behaviour in school settings often reinforces these unhealthy eating patterns, as students may feel pressured to conform to their friends' choices, further exacerbating the problem [1,3]. The availability of inexpensive, unhealthy food options around schools poses a considerable challenge to promoting better nutrition and overall health among young people. A school environment supportive of healthy eating is essential to combat heavy marketing of unhealthy food. Modification of the school food environment (including high level policy changes at state or national level) can have a positive impact on eating behaviours [14].

### 3.3. Estimation of caloric intake from school food vendors and its implications for nutritional status in schoolchildren

The study investigated the caloric intake of students from various age groups (5–8, 9–13, and 14–18 years) through food purchases from vendors around surveyed schools. The findings revealed key insights into the adequacy of school meals in meeting the daily nutritional requirements. Table 2 presents estimate of the amount of calories these students obtain from portions of different foods and beverages purchased from food vendors at their schools. The Table also shows the percentage preference for different types of foods available at the surveyed schools. According to the National School Feeding and Nutritional Services Guidelines [11], it is recommended that food served at school should provide 60–75% of the daily recommended nutritional requirements. The findings of the study indicate that the actual caloric intake from foods purchased at school varies by age group and the type of food consumed. Specifically, the percentage of the Recom-mended Daily Allowance (RDA) of calories obtained from foods purchased at the school food vendors were (Table 2):

• Age group 5–8 years: 8–24% of the daily recommended caloric intake.

• Age group 9–13 years: 6–19% of the daily recommended caloric intake.

• Age group 14–18 years: 5–16% of the daily recommended caloric intake.

**Table 1. Common food available around the school environment and their price.**

| Types of food | Price (TZS) (USD) | Description |
|---|---|---|
| Samosa | 100 (0.00037) | A fried or baked pastry with a spicy potato filling. The filling is wrapped in dough and folded into a triangular or cone-shaped |
| Fried cassava | 100 (0.00037) | Big cassava chips deep fried in cooking oil |
| kachori | 100 (0.00037) | Fried mashed potato balls covered in wheat flour |
| Mandazi | 100 (0.00037) | A form of fried bread (deep fried) |
| Potato Chips | 500−1000 (0.183-0.37) | Fresh Potato chips fried in vegetable oil normally go with chicken or fish but in the visited schools it is just fried potato chips |
| Bajia | 100 (0.00037) | Deep fried ball of wheat flour and beans or pigeon peas and spices (garlic, parsley, curry powder, cumin, red pepper, hot pepper, onions, baking powder |
| Fried sweet potato | 100 (0.00037) | Same as cassava fried chips |
| Half cake | 100 (0.00037) | Deep fried rolled ball of wheat fluor, sugar, baking powder, unsalted butter, egg, milk etc. It is a deep-fried product with crunchy crust made from chemically leavened wheat flour dough |
| Vishet | 100(0.00037) | coconut pulp, sugar plus food colours |
| Rice with beans | 100−1000 ((0.00037)-0.37) | One measure (i.e., one spoon is TZS 100, thus one student can one or more spoons up to a maximum of 10 spoons equals to TZS 1000 |
| Bread | 100 (0.00037) | Baked and leavened food made of a mixture of flour (wheat), water and yeast |
| Mango | 100 (0.00037) | Fresh mango fruit or slices |
| Fruit juice | 200 (0.00074) | Fresh cocktail Juice, single fruit, sold in a glass of approximately 250 ml |
| Fried Irish potatoes | 100 (0.00037) | Large pieces of Irish potato deep fried |
| Ice cream | 100 (0.00037) | Frozen dessert made from milk or cream flavoured with sugar |
| Pilau | 1000 (0.0037) | A rice dish cooked in broth with meat and spices. Sold on small plates approximately 150g |
| Candy | 100−200 (0.00037-0.00074) | Sweet confectionary |
| Biscuits | 100−600 (0.00037-0.220) | Small baked unleavened cake, typically flat crisp and sweet |
| Carbonated drinks | 200−600 (0.00074-0.220) | Contain small bubbles of $CO_2$ such colas and others |
| Salted snacks | 300−500 (0.110-0.183) | Snacks made from sliced whole potatoes or nuts containing a lot of salt |

NB: Number. in bracket＝USD cents.

Across all age groups, the majority of calories were obtained from deep-fried potato chips (16–24% of the RDA) and cooked rice (13–19% of the RDA), which were the most frequently purchased items from the school food vendors.

The study highlights a significant gap in the adequacy of the calories students receive from school meals. According to the national guidelines, meals provided at schools should ideally contribute 60–75% of students' total daily caloric intake. However, the data reveals that the actual contribution of school meals to students' caloric needs is significantly lower, as shown above. This discrepancy suggests that school meals may not be supplying sufficient energy to support optimal growth, development, and performance in schoolchildren and adolescents. Insufficient caloric intake in schoolchildren and

**Table 2. Amount of calories student get from foods served at school compared to RDA.**

| Type of food most preferred by school children | Potato samosas | Fried cassava | Kachori | Chips | Cooked rice and beans | Fruit juice |
|---|---|---|---|---|---|---|
| Percentage (foods preferred by students) | 95.70% | 87% | 73.90% | 21.70% | 43.50% | 17.40% |
| Unit of measurement (g) | Piece | Piece | Piece | Small plate | Small plate | Glass (ml) |
| **Estimated weight g** | 50 | 30 | 25 | 150 | 150 | 250 |
| Energy Kcal/100 g as stipulated in Tanzania food composition Table | 194 | 346.5 | 237.6 | 249 | 199.1 | 45 |
| Energy Kcal/weight consumed by each school child | 97 | 103.9 | 59.2 | 373.5 | 298.6 | 11.2 |
| Assuming that a child buys 2 pieces of snack or one plate of rice plus a glass of juice | 205.2 | 219 | 129.6 | 384.7 | 309.8 | |
| RDA for school children Aged 5–8 years (Kcal) | 1600 | 1600 | 1600 | 1600 | 1600 | 1600 |
| RDA for school children Aged 9–13 years (Kcal) | 2000 | 2000 | 2000 | 2000 | 2000 | 2000 |
| RDA for school children Aged 14–18 (Kcal) | 2400 | 2400 | 2400 | 2400 | 2400 | 2400 |
| Day school food served should meet 60–75% of 1600 RDA (Kcal) for age 5–8 years | 960 −1200 | 960-1200 | 960-1200 | 960-1200 | 960-1200 | 960-1200 |
| Day school food served should meet 60–75% of 2000 RDA (Kcal) for age 9–13 years | 1200 −1500 | 1200-1500 | 1200- 1500 | 1200-1500 | 1200 −1500 | 1200 −1500 |
| Day school food served should meet 60–75% of 2400 RDA (Kcal) for age 14–18 years | 1440-1800 | 1440-1800 | 1440-1800 | 1440-1800 | 1440-1800 | 1400 −1800 |
| **Percentage of average RDA each child get from meals served at School assuming that a child buy at least 2 pieces of snack and a glass of juice** | | | | | | |
| RDA for school children Aged 5–8 years (Kcal) | 13 | 14 | 8 | 24 | 19 | |
| RDA for school children Aged 9–13 years (Kcal) | 10 | 11 | 6 | 19 | 15 | |
| RDA for school children Aged 14–18 years (Kcal) | 9 | 9 | 5 | 16 | 13 | |

adolescents has well-documented effects on both physical health and academic performance. Malnutrition, particularly caloric insufficiency, is a significant contributor to stunting (low height-for-age), a condition that has been reported to affect a significant proportion of schoolchildren in Tanzania. A recent cross-sectional survey conducted in Tanzania Mainland reported a 32% prevalence of stunting among school adolescents. Moreover, a study by John et al. (2021) found an even higher prevalence of stunting (45.8%) among older adolescents (15–19 years). Stunting is often accompanied by deficiencies in key micronutrients, such as iron deficiency, which was observed in 34% of the surveyed adolescents. These deficiencies can result in cognitive impairments, reduced physical performance, and increased susceptibility to infections, which in turn affect students' academic achievement and behavioral health. The lack of adequate nutrients also compromises brain function, leading to difficulties in concentration, memory, and problem-solving, all of which are essential for academic success. Moreover, inadequate nutrition can have long-term consequences on emotional well-being, contributing to mood swings, irritability, and increased risk of mental health disorders.

Given the findings of this study and the prevalent nutritional deficiencies observed among schoolchildren and adolescents, there is an urgent need for policy reforms, strategies, and interventions aimed at improving the nutritional quality of school meals. Policies should ensure that school food vendors are required to provide foods that meet the nutritional standards outlined in the National School Feeding and Nutritional Services Guidelines. This would include offering more nutrient-dense foods, such as fruits, vegetables, legumes, and lean proteins, in addition to reducing the availability of high-calorie, low-nutrient foods like deep-fried snacks and refined carbohydrates.

The implications of malnutrition among schoolchildren and adolescents extend beyond immediate health outcomes. As future parents and labor force participants, children and adolescents represent a substantial portion of the population that will drive the nation's economic development and socioeconomic stability. According to the National Census 2022, children and adolescents (ages 5–18) make up 37.6% of the total population, equating to approximately 23,198,206

individuals. Therefore, ensuring that these students receive adequate nutrition is not only crucial for their individual health and development but also for the broader national health and economic goals.

The study underscores the importance of addressing nutritional deficits in schoolchildren and adolescents, particularly the insufficient caloric intake from meals provided by school food vendors. Given the high prevalence of stunting and micronutrient deficiencies in Tanzania, there is a pressing need for nutritional interventions within the school environment to support optimal growth, cognitive development, and academic performance. By aligning school feeding programs with national guidelines and improving the quality of foods provided at schools, it is possible to reduce the risks associated with malnutrition and promote better health outcomes for future generations.

### 3.4. Healthiness of foods and beverages within and around the school

A healthy school food environment encourages students to make better food choices. According to the NOVA Food Classification system [6,15], foods are categorized based on the processes they undergo after being separated from nature and before consumption. The classifications include: unprocessed or minimally processed, processed culinary ingredients, processed, and ultra-processed. An assessment of the foods and beverages sold around schools at Temeke Municipal, Tanzania, indicated that 11.11% fall under the category of unprocessed or minimally processed and 11.11% processed culinary ingredients), 33.33% processed foods while 44.44% were ultra-processed (Table 3). The majority of foods found in the school environments were consisting of items high in calories, added sugars, saturated or trans fats, and sodium, but low in essential nutrients [16]. These unhealthy dietary patterns raise significant public health concerns, contributing to obesity and chronic diseases among school-aged children [17].

Research shows that many children consume high amounts of fried foods, sweets, and sugary drinks. For example, a survey in Tanzania demonstrated high intakes of junk food among school-aged children [18]. Similarly, a study in Kenya found that most primary school children consumed sweetened beverages and junk foods, including chips, sweets, sausages, doughnuts, and chocolate, in the week prior to assessment [19]. There is growing evidence that children in developing countries are increasingly making unhealthy food choices, largely due to a lack of knowledge and misconceptions about nutrition [12]. This trend reflects a shift in the perception of food, evolving from a source of nourishment to a lifestyle marker and source of pleasure, often influenced by media that promotes high consumption of processed foods rich in calories, fat, and sugar, while offering little to no essential nutrients [20]. To combat rising obesity rates and associated health risks among school-aged children, it is crucial to empower them to make healthier food choices through nutritional education and by promoting positive attitudes toward healthy eating.

### 3.5. The presence of food products in the school environment: Implications for the nutrition and health of school children and adolescents

The presence and consumption of specific food products sold by food vendors around schools have a profound impact on the nutritional status and overall health of schoolchildren and adolescents [21]. These foods, which include a combination of highly processed snacks, sugary beverages, and some more nutritious options, can significantly influence students' dietary habits, growth, and academic performance. [21]. The categories of foods sold by vendors at the surveyed schools are briefly discussed respecting their nutritional implications (Fig 2).

(i) **Candies, sweet snacks, and deep-fried snacks (100% Availability)**

- **Candies and sweet snacks**: These items are typically high in sugars, fats, and calories while being low in essential nutrients like vitamins, minerals, and fiber. Excessive consumption of sugary foods can contribute to the development of obesity, dental caries, and insulin resistance, particularly in schoolchildren and adolescents. A study conducted in Nairobi, Kenya established that there is a significant relationship between energy-dense foods found around schools and the nutritional status of the adolescents [22]. Over time, regular intake of these foods can

**Table 3. Classification of food available at schools assessed at Temeke district.**

| Type of food | Ingredients/Description | Unprocessed/ Minimally processed | Processed culinary ingredients | Proceed foods | Ultra- processed foods |
|---|---|---|---|---|---|
| Samosa | A small triangular pastry (wheat flour) filled with spiced meat or vegetables and deep fried in oil. | | | | ☑ |
| Kachori | A spicy, deep-fried snack that is often filled with a mixture of lentils or potatoes, and spices and deep **fried** | | | | ☑ |
| Fried Cassava | Cassava is boiled then cut into large chips and deep fried in oil eaten with sprin-kled salt and chill and **deep fried** | | | | ☑ |
| Irish Potato Chips | A thin slice of potato fried in oil and salted or flavored and **deep fried** | | | | ☑ |
| Mandazi | Wheat dough deep fried in oil, ingredients include water, wheat flour, yeast, sugar, **deep fried** | | | | ☑ |
| Half cake | Wheat flour dough with sugar, baking soda and baked | | | ☑ | |
| Bagia | Bean cake, sugar added, baked | | | | ☑ |
| visheti | coconut pulp, sugar plus food colors and **deep fried** | | | | ☑ |
| Sweet potato | Sweet potato slice **deep fried** | | | | ☑ |
| Bread | A mixture of flour, water, salt, yeast and other ingredients, kneaded until the flour is converted into a stiff paste or dough, followed by baking the dough into a loaf, salt added | | | ☑ | |
| Tambi (spaghetti) | Long, thin, solid cylindrical paste produced by mixing milled wheat, water, eggs (sometimes optional ingredients). Ingredients are thoroughly mixed and extruded. Can be boiled and eaten with stew | | | ☑ | |
| Cooked rice | Also referred to as boiled rice is usually cooked on high heat until a rolling boil, then simmered with the lid on, and steamed over the residual heat after turning off the heat. It is usually served with stew of beans or meat or fish or vegetables | | | ☑ | |
| Mangoes | Fresh fruits are eaten raw | ☑ | | | |
| Fruit Juice | Fresh Juice | | | ☑ | |
| Pilau | A dish of rice, meat etc. seasoned with spices and cooked in a broth or baked | | | ☑ | |
| Cowpea | Fruits | ☑ | | | |
| Meat Soup | Meat **boiled, salt added,** spices added | | ☑ | | |
| Cooked beans | **boiled,** species and **salt added** | | ☑ | | |
| | Percentage (%) | 11.11 | 11.11 | 33.33 | 44.44 |

increase the risk of metabolic disorders [23] like type 2 diabetes and contribute to the rise of non-communicable diseases (NCDs).

- **Deep-fried cassava chips, deep-fried sweet potatoes, and deep-fried irish potatoes**: Fried snacks are energy-dense and high in trans fats or saturated fats, which can increase the risk of cardiovascular diseases, obesity, and dyslipidemia (imbalanced levels of fats in the blood). Although they provide calories, they offer little nutritional benefit in terms of vitamins and minerals. The high glycemic index of these foods can cause blood sugar spikes, leading to energy crashes that may impair concentration and focus in the classroom.

(ii) **2. Samosas (95.7% Availability)**

- Samosas are typically made with refined flour, saturated fats, and sometimes meat fillings. While they provide some protein from the meat, they are primarily a source of empty calories. The high fat content and the use of

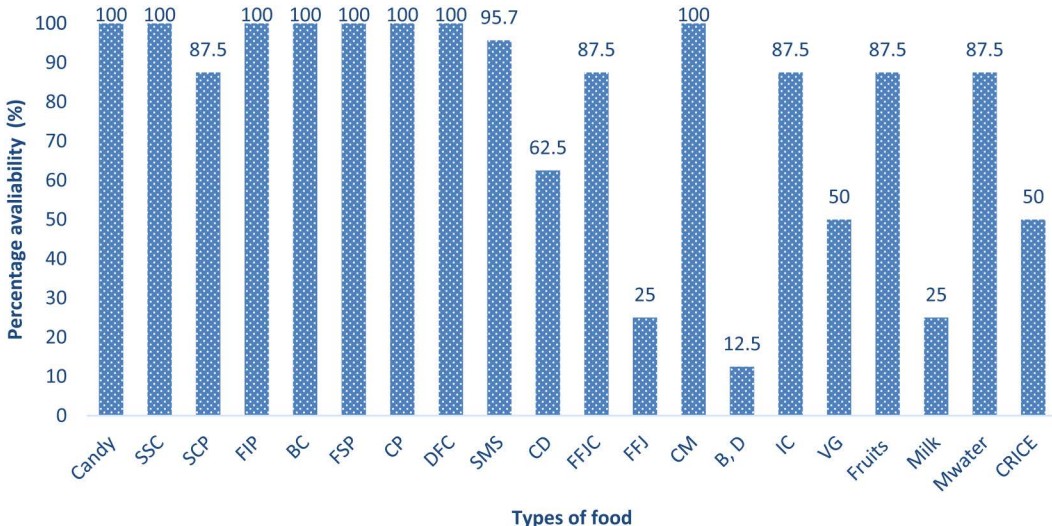

**Fig 2. Food available in surveyed schools.** NB: SSC = Sweet Snacks, SCPS = Salted Crips, CD = Carbonated drinks, FFJC Fresh Fruit Juice with sugar added, FFJ = Fresh Fruit Juice no sugar added, CM = cooked meal, B, D Bread, Donuts, IC = Ice cream, VG = Vegetables, Fruits = Mixed Fruits, CRICE = Cooked rice, M. water = Mineral water.

refined flour make them less nutrient-dense. Consuming samosas frequently can contribute to poor weight management and increase the risk of obesity and related metabolic disorders. Additionally, the high fat and carbohydrate content can lead to insulin resistance, a precursor to type 2 diabetes.

(iii) **Salted Crisps, Ice Cream, and Fresh Fruit Juice with Added Sugar (87.5% Availability)**

- Salted Crisps: are typically high in sodium and unhealthy fats, which can contribute to high blood pressure (hypertension) and cardiovascular diseases. Excessive salt intake is associated with an increased risk of stroke, kidney disease, and obesity. Additionally, their low fiber and micronutrient content make them poor choices for growing children who require nutrient-dense foods for development.

- Ice Cream: is high in sugars, saturated fats, and calories. Regular consumption can contribute to weight gain and increase the risk of insulin resistance. Ice cream also offers little in terms of essential nutrients like vitamins and minerals, and the sugar content can have a detrimental effect on oral health, contributing to tooth decay.

- Fresh Fruit Juice with Sugar Added: While fresh fruit juice contains some vitamins and minerals, the addition of sugar increases the glycemic load and calorie content, making it similar to soda in terms of its effects on blood sugar levels. Regular consumption can lead to weight gain, obesity, and an increased risk of type 2 diabetes. Moreover, the lack of fiber in fruit juice (due to the removal of pulp) means that it does not offer the same digestive benefits as whole fruits.

(iv) **Fruits and Mineral Water (87.5% Availability)**

- Fresh fruits are an excellent source of vitamins, minerals, fiber, and antioxidants. They provide essential nutrients like vitamin C, potassium, and folate, which are crucial for growth, immune function, and overall health. Fruits help in maintaining a healthy digestive system, skin health, and cognitive function. Including fruits in the school diet is vital for students' physical and mental development, as they help prevent nutrient deficiencies and chronic diseases.

- Mineral Water: Adequate hydration is critical for maintaining cognitive function, physical performance, and overall health. While mineral water is a healthy beverage choice, it's important that students choose water over sugary beverages to avoid the adverse effects of high-calorie, high-sugar drinks. Mineral water also provides essential electrolytes, which are vital for maintaining fluid balance and muscle function.

(v) **Carbonated drinks (65.2% Availability)**

- Carbonated drinks (Soda, Soft Drinks): These drinks are high in sugars (often in the form of high-fructose corn syrup) and provide empty calories with no nutritional benefit. They are one of the leading contributors to obesity, tooth decay, and the development of insulin resistance. The excessive sugar content can lead to spikes in blood glucose levels, followed by crashes that affect concentration and learning. In the long term, frequent consumption of sugary carbonated drinks is linked to an increased risk of metabolic syndrome, type 2 diabetes, and cardiovascular diseases.

(vi) **Cooked Rice and Vegetables (50% Availability)**

- Cooked rice: Rice, especially white rice, is a refined carbohydrate and has a high glycemic index, meaning it can lead to rapid spikes in blood sugar levels. Although it provides energy in the form of carbohydrates, rice lacks essential vitamins, minerals, and fiber. Pairing rice with vegetables or proteins can make it more nutritionally balanced. However, frequent consumption of rice without adequate vegetables or protein can contribute to nutritional imbalances.

- Vegetables: When included in school meals, vegetables are an important source of fiber, vitamins, and minerals, such as vitamin A, vitamin C, and potassium. These nutrients are essential for immune function, cellular repair, and bone health. Increasing the availability of vegetables in school meals is crucial for improving the overall nutritional quality of students' diets.

(vii) **Milk and Fruit Juice with No Sugar Added (25% Availability)**

- Milk is a rich source of calcium, protein, and vitamin D, all of which are essential for bone health, muscle development, and immune function. Including milk in school meals can help meet the nutritional needs of growing children and adolescents. However, the milk provided is low-fat or skimmed to avoid excessive saturated fat intake.

- Fruit Juice with No Sugar Added: This is a better alternative to sugary fruit juices. Unsweetened fruit juice retains some of the vitamins, minerals, and antioxidants from the fruit, particularly vitamin C and potassium, without the added sugar. However, consuming whole fruits is preferable due to the additional fiber they provide, which is lost during juicing.

(viii) **Bread and doughnuts (12.5% Availability)**

- Bread: Depending on the type (e.g., whole-grain vs. refined), bread can provide carbohydrates and fiber. Whole-grain bread is a better choice because it provides more fiber, B vitamins, and minerals compared to refined bread. However, excessive consumption of refined carbohydrates (like white bread) can contribute to insulin resistance and weight gain.

- Doughnuts are high in sugar and saturated fats, which can contribute to weight gain, insulin resistance, and the development of obesity and cardiovascular diseases. They are low in nutritional value, providing few essential vitamins, minerals, or fiber.

The variety of food products sold by vendors around schools, especially those high in sugars, refined fats, and empty calories, poses significant challenges for maintaining optimal nutritional health in schoolchildren and adolescents. The

prevalence of deep-fried snacks, sugary beverages, and processed foods in the school environment increases the risk of obesity, insulin resistance, poor academic performance, and the development of chronic diseases like type 2 diabetes and cardiovascular diseases. To improve the nutritional status of students, it is essential to restrict the availability of unhealthy, nutrient-poor foods and promote the inclusion of nutrient-dense options such as fruits, vegetables, whole grains, and lean proteins in school food offerings.

### 3.6. Food preferences of school children

The survey findings revealed that most preferred foods by school children at Temeke Municipality were largely the deep-fried snacks, including samosa (95.5%) fried potato chips (87%) and fried marched potato bolls (73.9%) (Fig 3). Deep-frying is a cooking method that involves immersing food in oil at temperatures ranging from approximately 150–200°C, the process causes the cell walls in the food to rupture and form pores that facilitate oil absorption [24]. The absorption of oil in fried food increases with higher polyunsaturated fatty acids (PUFA) content in oils, lower frying temperatures, longer exposure times [24]. Frequent consumption of fried foods (four or more times a week) is associated with an increased risk of developing type 2 diabetes, heart failure, obesity, and hypertension, it also promotes the development of non-communicable diseases (NCDs) and raises mortality rates [25]. Deep fried foods are preferred due to their superior sensory qualities especially colour, odour, flavour, and texture. The appeal of deep-fried foods rises from their reasonable pricing, delicious taste, and the convenience of their quick and easy preparation process [18]. However, these foods are unhealthy because the frying process changes the chemical structure of fats from Cis to trans fats, and makes it difficult for the body to break it down, leading to negative health effects. Trans fats are associated with an increased risk of various diseases including heart related diseases, cancer, diabetes and obesity. The level of trans fats in the final deep-fried product depends on the type of oil used, temperature reached and frequency of reuse of the oil. In this case the common oil used is sunflower which has a short frying life and shelf life of the food product due to their susceptibility to oxidation and contains 20% TFA [26] A Study conducted in Morogoro municipal revealed that junk foods were mostly consumed by school children and were preferred [27]. Thus, the food industry needs to develop safe products and at the same time accommodate the consumer's tests and preferences. This study was conducted in a view to provide insight onto the target consumers food preference in developing a nutrient dense snack bar for school children and adolescents in a view to tackle the challenge of micronutrient deficiency among this group.

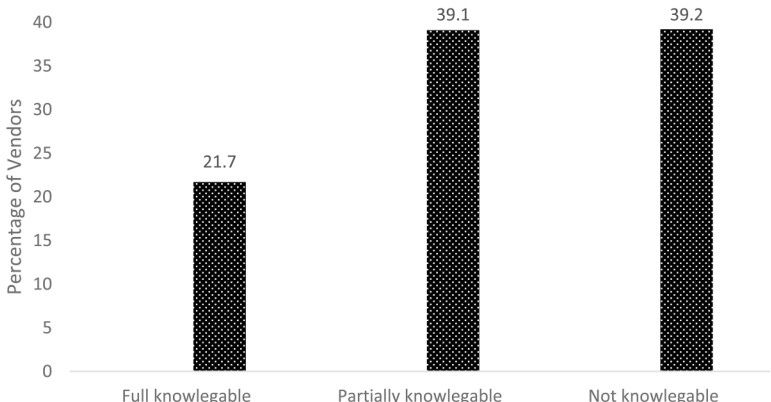

**Fig 3. Types of food found around the environment.** NB: SMS = samosa, FC Fried potato Chips, FMPB = Fried Marched potato bolls, BC = Bean cake, FSP = Fried Sweet Potato, FCP = Fried coconut pulp (with sugar and food colors), FJC = Mixed Fresh Juice, Fried Irish Potato.

### 3.7. Food vendor premises: Cleanness, hygiene and food safety

According to the Food regulation 2009 food vendors around or within schools fall under persons who prepare, pack or serve food and some parts of the regulation apply to them. Under section 36 parts that apply to school food venders were identified which guide them. Assessment of compliance of seven minimum requirements of food hygiene regulation by the food vendors is given in Table 4. From this assessment compliance to the minimum requirements of food hygiene regulation 2009 by food vendors are only 29%, whereas non-compliance is 71%.

The low level of compliance raises serious concerns regarding food safety and student health. Non-compliance can lead to increased risks of foodborne illnesses, which can have severe consequences, particularly for children whose immune systems are still developing. The presence of unsafe food handling practices in schools can contribute to out-breaks, potentially leading to significant health crises. Given that a large percentage of vendors are not adhering to hygiene standards, there is a need for comprehensive training programs. The programs should focus on educating vendors about the importance of food hygiene, safe food handling practices, and the specific requirements outlined in the Food Hygiene Regulation 2009. Training could be supplemented with practical workshops to ensure that vendors are equipped to comply with the regulations.

### 3.8. Food vendors' knowledge on food safety and nutrition

Food hygiene and safety involves practices that prevent contamination and minimize the risk of foodborne illnesses. The survey assessed the level of knowledge among food vendors about food safety and nutrition, findings from the survey revealed that among food vendors formally registered at school, 22% have fully knowledge on food safety and nutrition, 39% have partial knowledge and 39% had no knowledge indicated in Fig 4. Food vendors normally either prepare the food at the school or bring to school foods prepared to home and save to students. Knowledge of food safety and nutrition especially food safety hygiene to food vendors is critical to protect the health of the consumers.

It is very important to consider knowledge of food safety and nutrition for the selection and recruitment of food vendors to ensure they recognize their obligation to promote the health and safety of students. Food vendors must be equipped to prepare and serve healthy, balanced diets, which requires an assessment of their knowledge regarding food preparation, balanced meals and food groups. In this study, an evaluation of vendors' understanding revealed that only 30% (n = 20) could identify four out of five food groups that constitute a balanced diet. Meanwhile, 56.5% (n = 20) could identify two to three groups, and 30% (n = 20) could not identify any food groups at all. Based on the findings, schools and relevant authorities should consider conducting training sessions focusing on food safety practices and nutrition education for

**Table 4. Compliance with hygiene regulation.**

| S/N | Minimum food hygiene regulation requirement | Complied | Not complied |
|---|---|---|---|
| 1 | Not use his bare hands to handle any unwrapped food | | 1 |
| 2 | Take all reasonable steps to prevent customers from using their bare hands to handle any unwrapped food | | 1 |
| 3 | Not use his breath to open any bag or wrapper intended for use in such preparation, packing and serving; | | 1 |
| 4 | Wipe his hands with clean towel or any other clean and suitable material | 1 | |
| 5 | Avoid the placing, carrying or storing any unwrapped food in such a manner that a plate, dish or container comes into contact with the food in other plate, dish or container | | 1 |
| 6 | Not use any raw material or ingredient that may be contaminated with parasites, pathogenic microorganisms, toxic, or decomposed or foreign substances which may cause the finished food product unfit for human consumption | 1 | |
| 7 | Not keep any raw material, ingredient, intermediate food product or finished food product at temperatures that is likely to support the reproduction of pathogenic microorganisms or the formation of toxins | | 1 |
| | Total | 2 | 5 |
| | Percentage (%) compliance | 29 | 71 |

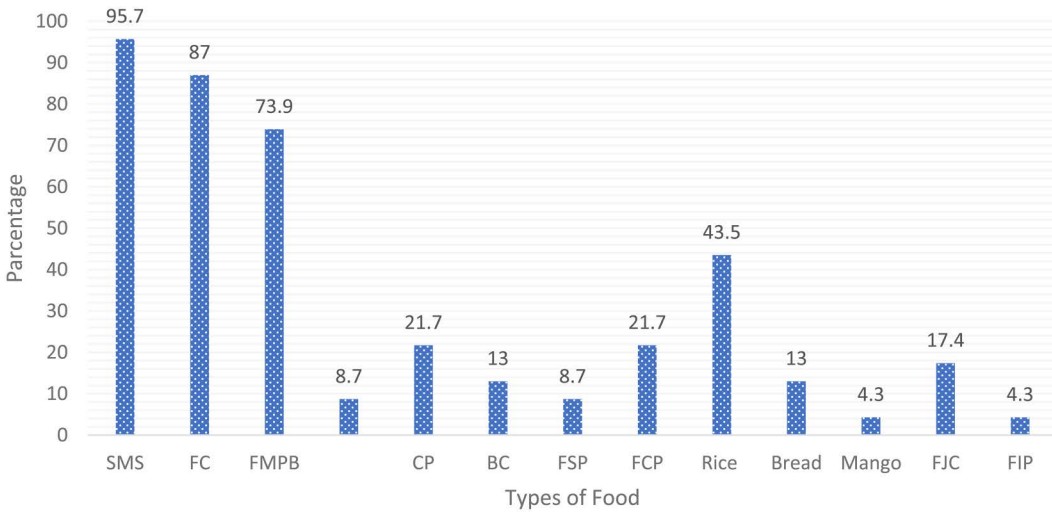

**Fig 4. Vendor knowledge on food safety and nutrition.**

enhancing food vendor's knowledge. Regular assessments and certifications could also help maintain high standards and ensure that all vendors are equipped to provide healthy and safe meals.

## 4. A critical review of the literature on the school feeding policy framework

The review unravelled strengths, gaps and challenges and came up with concrete recommendations to address the challenges and close the policy gaps. Table 5 summarizes the outcomes of this review.

## 5. Conclusion and recommendations

The survey findings highlight the school food environment exposes students to unhealthy options, particularly deep-fried foods high in sugars and salts, while also compromising food safety, indicating a gaps in the school food environment, particularly the limited availability of healthy food options. Given that the current food environment offering predominantly energy-dense, nutrient-poor foods, there is a clear need for interventions that prioritize healthier choices for school-aged children. One potential solution is the development of a snack bar designed to meet at least 75% of the daily nutritional requirements for school children. The snack bar would offer a structured, accessible means for students to consume nutrient-dense foods, complementing their overall dietary needs. Given the findings of this study, a snack bar would not only provide a healthier alternative to the current food environment but also address barriers such as affordability and convenience.

**Limitation of the study**

1. The study was conducted in a single district (Temeke District) in Dar es Salaam, which may limit the generalizability of the findings to other regions in Tanzania or to countries with different food environments. The sample may not fully represent the diverse socioeconomic backgrounds or regional variations in food access, potentially affecting the broader applicability of the results. A comprehensive study covering several regions will be done during scale up.

2. The study did not collect qualitative data, which could have provided deeper insights into the attitudes, perceptions, and experiences of students, school staff, and food vendors. The qualitative data could capture information on the complexities of students' food preferences or the social and cultural factors that influence dietary choices.

**Table 5. Strengths, policy gaps, challenges, and recommendations for school feeding programs in Tanzania.**

| Strength | Policy gaps and challenges | Recommendations |
|---|---|---|
| **Policy alignment and focus**: The Tanzanian government has recognized the importance of school feeding for improving student attendance, academic performance, and reducing malnutrition. The National Development Vision 2025 and the 2014 Education and Training Policy emphasize the provision of nutritious meals, safe water, and electricity in schools [28] | Tanzania lacks a unified and clear policy on school feeding, and there are no standardized guidelines for school meal quality. Participation in school feeding programs is not mandatory, leaving many students hungry [29]. | **Develop clear school feeding policies**: Formulate a clear and comprehensive national policy on school feeding that ensures uniformity in meal quality and compulsory participation in feeding programs [30]. |
| **Evidence of positive impact**: Successful school feeding programs, particularly in areas like Arumeru and Babati, have shown that local approaches can improve attendance, reduce absenteeism, and enhance academic performance [31]. | **Inconsistent funding**: School feeding programs often face financial constraints, with limited funds allocated by the government. Dependence on donors for funding also affects the sustainability of these programs [7,32]. | **Secure sustainable funding**: The government should allocate consistent and sufficient funding for school feeding programs and explore partnerships with local communities and private stakeholders to ensure sustainability [33]. |
| **Involvement of key stakeholders**: The involvement of MoEST, local governments, communities, parents, and development partners is recognized as key to the success of school feeding initiatives [34]. | **Limited community engagement**: While communities and parents are acknowledged, their actual involvement is minimal. Many parents view the fee-free education policy with misconceptions, affecting their participation in the program [33]. | **Increase community and parental involvement**: Strengthen community and parental engagement through awareness campaigns, training, and incentivizing participation in school feeding programs [33]. |
| **Impact on student health and attendance**: School meals are linked to improved student health, reduced absenteeism, and better concentration [35]. | **Inadequate infrastructure**: Many schools lack proper facilities such as functioning kitchens, dining halls, and food storage. This reduces the quality and safety of meals [36]. | **Invest in school infrastructure**: Prioritize investments in building and upgrading school infrastructure, including kitchens, dining halls, and storage facilities, to support the effective implementation of school feeding programs [37]. |
| **Promotion of healthy eating**: The provision of school meals can help students form healthier dietary habits and improve nutrition [38]. | **Unhealthy food environment**: In some schools, the food sold by vendors is unhealthy (e.g., deep-fried snacks), exposing students to poor nutrition despite school meal initiatives [8]. | **Regulate and monitor food vendors**: Establish strict guidelines for food vendors within and around schools. Ensure that the food sold is nutritious and aligns with national food guidelines. Regular monitoring and health checks for vendors are also necessary [36]. |
| **Policy documentation and guidance**: There are documents outlining the basic elements of school health and nutrition services, which define roles and responsibilities for various stakeholders [38]. | **Absence of a comprehensive implementation framework**: While policies and strategies exist, there is a lack of a comprehensive framework for tracking and implementing the activities outlined, especially concerning vulnerable groups [39] . | **Create a robust implementation framework**: Develop a detailed, actionable framework for the implementation, monitoring, and evaluation of school feeding programs, with a focus on vulnerable groups (e.g., orphans, children with disabilities) [40]. |
| **Local examples of success**: Programs by the World Food Programme and Project Concern International show successful models of home-grown school feeding programs [41,42]. | **Limited scaling of successful models**: Successful local school feeding programs are not scaled out to other regions, limiting their impact and reach [33]. | **Scale up home-grown models**: Learn from successful home-grown school feeding models and scale them up across the country, particularly by integrating local food production and community involvement [36]. |
| **Emphasis on multisectoral approach**: There is a recognition of the need for a multisectoral approach, involving sectors like agriculture, health, and education [43]. | **Weak intersectoral coordination**: Despite the policy's emphasis on multisectoral involvement, poor coordination among relevant sectors hampers the effective execution of school feeding programs[44]. | **Strengthen intersectoral coordination**: Improve collaboration and coordination among sectors such as education, health, agriculture, and local governments to ensure the smooth execution of school feeding initiatives [41]. |
| **Policy alignment with national development goals**: School feeding policies align with the national goals of reducing poverty and malnutrition, contributing to the country's development vision [45] . | **Absence of age-appropriate strategies**: Policies do not fully recognize or address the varied nutritional needs of children at different stages (e.g., early childhood and adolescence) [46]. | **Implement age-specific strategies**: Develop policies that address the unique nutritional needs of children at different educational levels, from early childhood to secondary school [36]. |

## Author contributions

**Conceptualization:** Nyabasi Makori, Anselm P Moshi, Hoyce Mshida, Dyness Kejo, Beatrice Bachwenkizi, Devotha Mushumbusi, Zahara Daudi, Monica Chipungahelo, Ai Zhao.

**Data curation:** Nyabasi Makori, Anselm P Moshi, Hoyce Mshida, Dyness Kejo, Beatrice Bachwenkizi, Zahara Daudi, Monica Chipungahelo, Ai Zhao.

**Formal analysis:** Nyabasi Makori, Anselm P Moshi, Hoyce Mshida, Ai Zhao.

**Funding acquisition:** Nyabasi Makori, Anselm P Moshi, Hoyce Mshida, Dyness Kejo, Beatrice Bachwenkizi, Ai Zhao.

**Investigation:** Ai Zhao.

**Methodology:** Nyabasi Makori, Anselm P Moshi, Hoyce Mshida, Dyness Kejo, Beatrice Bachwenkizi, Devotha Mushumbusi, Zahara Daudi, Monica Chipungahelo.

**Project administration:** Ai Zhao.

**Resources:** Anselm P Moshi, Ai Zhao.

**Supervision:** Nyabasi Makori, Anselm P Moshi, Hoyce Mshida, Dyness Kejo, Devotha Mushumbusi, Ai Zhao.

**Writing – original draft:** Nyabasi Makori, Anselm P Moshi, Hoyce Mshida, Dyness Kejo, Beatrice Bachwenkizi, Devotha Mushumbusi, Zahara Daudi, Monica Chipungahelo, Ai Zhao.

**Writing – review & editing:** Nyabasi Makori, Anselm P Moshi, Hoyce Mshida, Dyness Kejo, Beatrice Bachwenkizi, Devotha Mushumbusi, Zahara Daudi, Monica Chipungahelo, Ai Zhao.

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
