## [Decision Letter · Decision Letter 0]

Dear Dr. Makori,

Thank you for submitting your manuscript to PLOS ONE. After careful consideration, we feel that it has merit but does not fully meet PLOS ONE’s publication criteria as it currently stands. Therefore, we invite you to submit a revised version of the manuscript that addresses the points raised during the review process.

We look forward to receiving your revised manuscript.

Kind regards,

António Raposo

Academic Editor

PLOS ONE

Journal Requirements:

2**.** We note that the grant information you provided in the ‘Funding Information’ and ‘Financial Disclosure’ sections do not match.

“We acknowledge funding from Bright future “

4. In the online submission form, you indicated that “Data cannot be shared publicly, is available upon request”

6. Please ensure that you refer to Figure 2 in your text as, if accepted, production will need this reference to link the reader to the figure.

Reviewers' comments:

Reviewer's Responses to Questions

**Comments to the Author**

1. Is the manuscript technically sound, and do the data support the conclusions?

Reviewer #1: Yes

Reviewer #2: Yes

Reviewer #3: Yes

2. Has the statistical analysis been performed appropriately and rigorously?

Reviewer #1: No

Reviewer #2: Yes

Reviewer #3: Yes

3. Have the authors made all data underlying the findings in their manuscript fully available?

Reviewer #1: Yes

Reviewer #2: Yes

Reviewer #3: Yes

4. Is the manuscript presented in an intelligible fashion and written in standard English?

Reviewer #1: Yes

Reviewer #2: Yes

Reviewer #3: Yes

Reviewer #1: I am grateful for the opportunity to review this study, which addresses a topic of great relevance to public health and child nutrition. Assessing the school food environment and its impact on the eating habits of school-age children is essential for formulating effective food security and nutrition policies. The manuscript presents a relevant approach, with empirical data and a pertinent discussion, but there are points that could be improved to strengthen the scientific quality of the article.

It is recommended to expand the final section of the abstract to emphasize how the findings can inform future policies and interventions.

The introduction contextualizes the problem well, addressing the influence of the school food environment on child health and mentioning the triple burden of malnutrition. However, there is a lack of more recent and robust references to support the arguments. In addition, the justification for the study could be strengthened with a more in-depth discussion of gaps in the literature and the specific relevance of the Tanzanian context.

The justification for the choice of the four schools could be more detailed. In addition, the statistical methodology employed in the data analysis is not sufficiently explained; it is recommended to include information on the statistical tests used and the criteria for statistical significance.

The discussion lacks a more critical approach to the study's limitations. The comparison with international studies is limited and could be expanded to better contextualize the results within the global panorama of child nutrition. In addition, the discussion could go into more detail about viable solutions for improving the school food environment in Tanzania.

It is suggested to add in the conclusion section practical guidelines for policy interventions and food education programs, as well as suggesting future studies to further explore the issues raised.

Reviewer #2: The study on the school food environment in Temeke Municipality provides insights into the dietary habits of school-age children and highlights several critical issues.

The finding that 62.5% of schools partially implemented feeding guidelines is concerning. The lack of full compliance suggests a systemic issue in prioritizing children's health and nutrition. The study could benefit from exploring the barriers to full implementation, such as resource limitations or lack of training.

The lack of dining halls and functional kitchens in most schools raises questions about the feasibility of promoting healthy eating. Dining facilities are essential for encouraging communal eating and improving the overall dining experience. The absence of these facilities suggests a neglect of the physical environment necessary for healthy eating practices.

While the paper reports that 55.6% of food and beverages are classified as healthy, the prevalence of popular unhealthy items, such as samosas and fried snacks, indicates a significant issue. The high consumption of unhealthy options raises concerns about the overall dietary quality available to students. A more detailed analysis of the types of healthy foods offered and their accessibility would strengthen the findings.

The low level of knowledge regarding food safety and nutrition among food vendors, with only 22.0% aware of national food guidelines, is alarming. This lack of awareness could lead to unsafe food practices and poor nutritional offerings. The work could be enhanced by recommending specific training programs for vendors to improve their understanding of food safety and nutrition.

The comparison of calorie portions with the Recommended Dietary Allowances (RDA) reveals a concerning trend of insufficient caloric intake among students. However, the study lacks a deeper exploration of how these low percentages affect the overall health and academic performance. It would be beneficial to assess the long-term implications of such dietary inadequacies.

While the work highlights the need for interventions, it does not propose specific strategies or frameworks for implementing changes. A more constructive approach would involve outlining potential interventions, such as policy changes at the school level, community engagement, or partnerships with health organizations.

The study is based on a small sample size across four schools. The findings may not be generalizable to other regions or municipalities. A broader study could provide more comprehensive data on school food environments across different contexts.

Reviewer #3: The manuscript is well written and comprehensive with significant findings

However, there are few comments for clarifications in the methodology part

What were the inclusion and exclusion criteria for the respondents?

How many vendors were involved in the pre-testing of interview instrument?

Any sample size calculation performed to achieve good power of study?

Any description and psychometric analysis of the instrument used?

**Do you want your identity to be public for this peer review?** For information about this choice, including consent withdrawal, please see our Privacy Policy

Reviewer #1: No

Reviewer #2: **Yes: ** M. João Lima

Reviewer #3: No

---

## [Author Response · Author response to Decision Letter 1]

28 May 2025

Authors responded carefully to all comments, refer the matrix response sheet attached.

---

## [Decision Letter · Decision Letter 1]

Assessing the School Food Environment and its Role on Healthy Eating Behaviors among School Age Children in Dar es Salaam, Tanzania

PONE-D-25-08254R1

Dear Dr. Makori,

We’re pleased to inform you that your manuscript has been judged scientifically suitable for publication and will be formally accepted for publication once it meets all outstanding technical requirements.

Kind regards,

António Raposo

Academic Editor

PLOS ONE

Additional Editor Comments (optional):

Reviewers' comments:

Reviewer's Responses to Questions

**Comments to the Author**

Reviewer #2: All comments have been addressed

2. Is the manuscript technically sound, and do the data support the conclusions?

Reviewer #2: Yes

3. Has the statistical analysis been performed appropriately and rigorously?

Reviewer #2: Yes

4. Have the authors made all data underlying the findings in their manuscript fully available?

Reviewer #2: Yes

5. Is the manuscript presented in an intelligible fashion and written in standard English?

Reviewer #2: Yes

Reviewer #2: This version of the article shows a significant improvement, as evidenced by more fluid writing, with some less dense and more explicit details. It is therefore publishable.

**Do you want your identity to be public for this peer review?** For information about this choice, including consent withdrawal, please see our Privacy Policy

Reviewer #2: **Yes: ** M. João Reis Lima

---

## [Editor Report · Acceptance letter]

PONE-D-25-08254R1

PLOS ONE

Dear Dr. Makori,

I'm pleased to inform you that your manuscript has been deemed suitable for publication in PLOS ONE. Congratulations! Your manuscript is now being handed over to our production team.

Kind regards,

on behalf of

Dr. António Raposo

Academic Editor

PLOS ONE